# Recent Advances of Sustainable Textile Fabric Coatings for UV Protection Properties

Nour F. Attia [1,*], Rokaya Osama [2], Sally E. A. Elashery [3,*], Abul Kalam [4,5], Abdullah G. Al-Sehemi [4,5] and Hamed Algarni [4,6]

1 Gas Analysis and Fire Safety Laboratory, Chemistry Division, National Institute for Standards, 136, Giza 12211, Egypt
2 Photometry and Radiometery Division, National Institute for Standards, 136, Giza 12211, Egypt
3 Chemistry Department, Faculty of Science, Cairo University, Gamaa Str., Giza 12613, Egypt
4 Research Center for Advanced Materials Science (RCAMS), King Khalid University, P.O. Box 9004, Abha 61413, Saudi Arabia
5 Department of Chemistry, College of Science, King Khalid University, P.O. Box 9004, Abha 61413, Saudi Arabia
6 Department of Physics, Faculty of Science, King Khalid University, P.O. Box 9004, Abha 61413, Saudi Arabia
* Correspondence: drnour2005@yahoo.com or nour.fathi@nis.sci.eg (N.F.A.); sally_elsayed_ahmed@cu.edu.eg (S.E.A.E.)

**Abstract:** The rapid progress in the use of textile fabric materials in various industrial and domestic applications requires the inclusion of smart functions to achieve comfortable and safety properties to the end users. However, among these functions is the protection against harmful UV rays that cause harmful effects to human beings and textile materials. To this end, coatings for smart textile fabrics have to be incorporated into textile fabrics. Therefore, in this review, recent advances in the development of coatings for sustainable textile fabrics for UV protection will be reviewed. Hence, the precursors, the synthesis routes and the types of coatings for sustainable textile fabrics will be reviewed. Furthermore, the UV protection action of the coatings for the protection of textile fabrics will be covered and studied. Interestingly, the multifunctional effect of the treated coatings, such as the antibacterial properties of the developed textile fabrics, will be also studied.

**Keywords:** coatings; sustainable coatings; textile fabric coatings; UV protection; graphene sheets; nanoparticles

## 1. Introduction

Increasing the dependence of utilizing textile-based materials in various industrial applications requires the incorporation of unusual properties to textile fabrics to accomplish the demands for the new usage [1]. Therefore, one of the demerits of textile fabric materials are their negative sensitivity against harmful UV rays; hence, it is compulsory to adapt textile fabrics for protection against the harmful effects of UV rays [2]. This adaption involves the incorporation of UV protection into the surface of the textile fabrics via surface treatments [3]. On the other hand, the Sun's optical spectrum daily emits rays consisting of infrared (IR), visible, and ultraviolet (UV) with a range of wavelengths starting from 200 nm and ending up at 1 mm [4]. However, UV radiation can be considered as the most detrimental component of the solar spectrum [5]. Basically, the UV rays are classified based on their wavelength; thus, there are three regions of UV rays based on their energy and wavelength, which are denoted as UV-C (200 nm–280 nm), UV-B (280 nm–315 nm), and UV-A (315 nm–400 nm) [6,7]. Interestingly, the negative impact of UV rays primarily depends on their region energy [4,8]. Fortunately, the UV-C region that is the most harmful region of UV rays for human beings is totally absorbed by the oxygen atoms forming the ozone layer [6,7,9], and hence its harmful effect is restricted by the stratosphere layer [4,8].

However, the UV-B region is hardly filtered by the stratosphere layer, and 5%–10% of its rays penetrate to the Earth as a consequence of the hole that exists in its ozone layer. Meanwhile, 90%–95% of UV-A rays are daily penetrating the Earth's atmosphere straightforwardly as a result of their high wavelength [7]. The UV radiation affects human beings and other living things negatively by inducing different physiological impacts with a plethora of acute and slow-rated consequences depending on the wavelength and energy [7]. UV-A is less energetic than UV-B; however, it can penetrate deeper through the second layer of the skin and the dermis, due to its longer wavelength [4,5] and hence it causes the generation of reactive oxygen species (ROS) [7] and reactive nitrogen species (RNS), which in turn alter the lipids, proteins, and DNA of the cells [5,6]. These oxidative damages are attributed to those highly reactive molecules and may result in wrinkle formation and cause aging of the skin, and in turn cause the development of skin cancer as a result of immunosuppression against infections [7], especially in the range from 360 nm to 380 nm [4]. Therefore, the incorporation of UV protecting agents over the surface of textile fabrics is a crucial step to prevent harmful effects on human beings and textile fabrics themselves due to their organic and natural origin [10]. Thus, the research trend for the development of textile fabric coatings against harmful UV rays has received much attention regarding several treatment routes utilizing different materials [11]. Additionally, UV ray protection was evaluated using a factor denoted as the Ultraviolet Protection Factor (UPF). Interestingly, the UPF values mainly measured in the range from 280–400 nm [8]. Recently, the fabrication of sustainable textile fabric coatings, derived mainly from renewable precursors, has received strong attention from researchers as a way to afford protection to textiles and human beings from serious effects of harmful UV rays [10]. This trend adheres to sustainable development goals (SDGs) that are primarily geared toward the valorization of renewable and abundant biomass wastes for high-tech applications affording sustainability and cost-effectiveness [12]. Hence, in this review article, the recent progress in the UV protection function of the coatings of textile fabrics is reviewed. Moreover, different nanocoating-based sustainable materials were briefly studied and classified. Additionally, the mechanistic actions of UV protection are discussed.

## 2. Textile Fabric Coatings for UV Protection

Various treatment routes have been implemented for coating the surface of textile fabrics in order to give the textile high protection against harmful UV rays [13–15]. Therefore, several types of textile fabric-based coatings, such as graphene, spherical nanoparticles, sustainable inorganic nanotubes, that were carried out using various treatment routes, such as immersion, electrophoretic deposition, vacuum filtration deposition, dip-coating and layer-by-layer techniques, will be studied. Among the sustainable materials, the graphene sheets were facilely synthesized from renewable precursors. Hence, this review will be focusing on the potential of the material-based coatings suggested for UV protection applications. Interestingly, the UPF value can be straightforwardly evaluated in the range from 280 nm up to 400 nm based on the Australian/New Zealand Standard (AS/NZS 4399:1996) according to the following equation:

$$\text{UPF} = \frac{\int_{280}^{400} E_\lambda \ S_\lambda \ d_\lambda}{\int_{280}^{400} E_\lambda \ S_\lambda \ T_\lambda \ d_\lambda}$$

where $E_\lambda$ refers to the relative erythemal spectral effectiveness (the action spectrum), $S_\lambda$ is the solar UV spectral irradiance for a typical summer period during midday in clear sky conditions, $d_\lambda$ indicates the increment in the wavelength, $\lambda$ represents the wavelength (nm), and $T_\lambda$ is the spectral transmittance of the specimen determined by the UV/Vis spectrophotometer equipment [16,17]. The uncertainty of measurement mainly depends on the accuracy and precision of the spectrophotometer used, as these vary from one to another.

### 2.1. Graphene-Based Coatings

Graphene (GRP) is a talented two-dimensional (2D) material with sp$^2$ carbon atom structure and has made breakthroughs in the field of material science once it was discovered and separated from graphite flakes [18,19]. The graphene is characterized by peculiar features such as outstanding electronic, electrical, thermal, chemical, mechanical and physical properties, and therefore it is promising in various industrial applications [20–27]. Various top-down and bottom-up synthesis routes have been implemented for the synthesis of graphene yielding graphene with a single layer and sheets with few layers of thickness (Figure 1) [28]. The detailed knowledge of different synthesis approaches of graphene has been gained and reported in our recent report [28]. Recently, the GRP sheets were prepared from renewable precursors such as biomass using water soluble polymers via green routes, as shown in Figure 2 (ultrasonication etc.) [28–32]. The GRP sheets derived from the recycling of sugar beet leaves as biomass was reported as an efficient UV filter and then displayed an excellent blocking efficiency of UV rays and meanwhile was transparent to visible rays, as shown in Figure 3 [28]. Hence, this affords a flexible, smart, and freestanding transparent UV protective filter (Figure 4) [28].

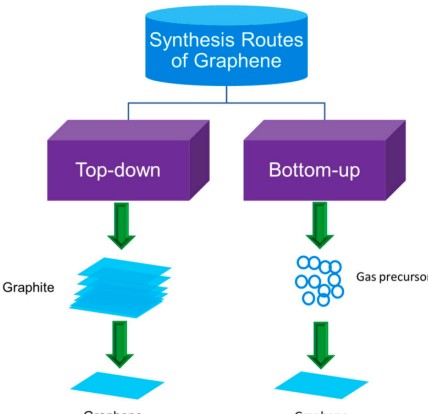

**Figure 1.** Schematic diagram representing the synthesis routes of graphene. Reproduced with permission [28]. Copyright 2021, Elsevier.

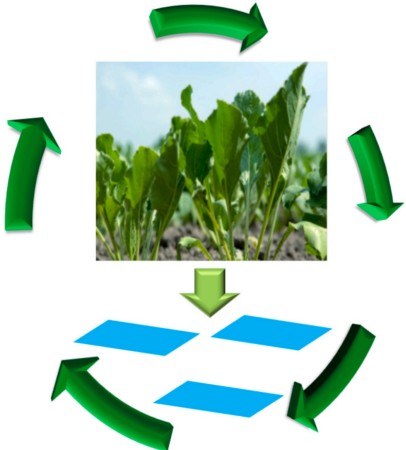

**Figure 2.** Schematic diagram representing the renewable circle of graphene sheets as new generation flame-retardant materials. Reproduced with permission [28]. Copyright 2021, Elsevier.

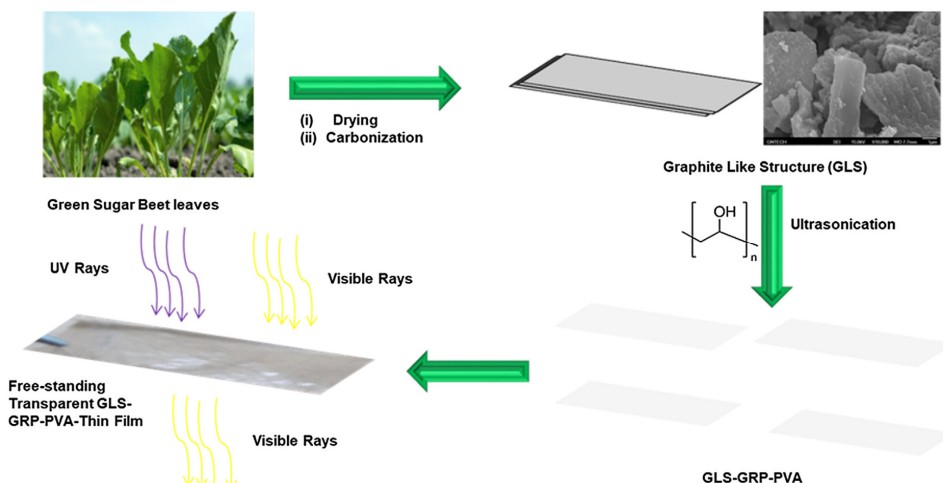

**Figure 3.** Schematic diagram representing the green synthesis of graphene sheets from sugar beet leaves in presence of PVA. Reproduced with permission [29]. Copyright 2021, Elsevier.

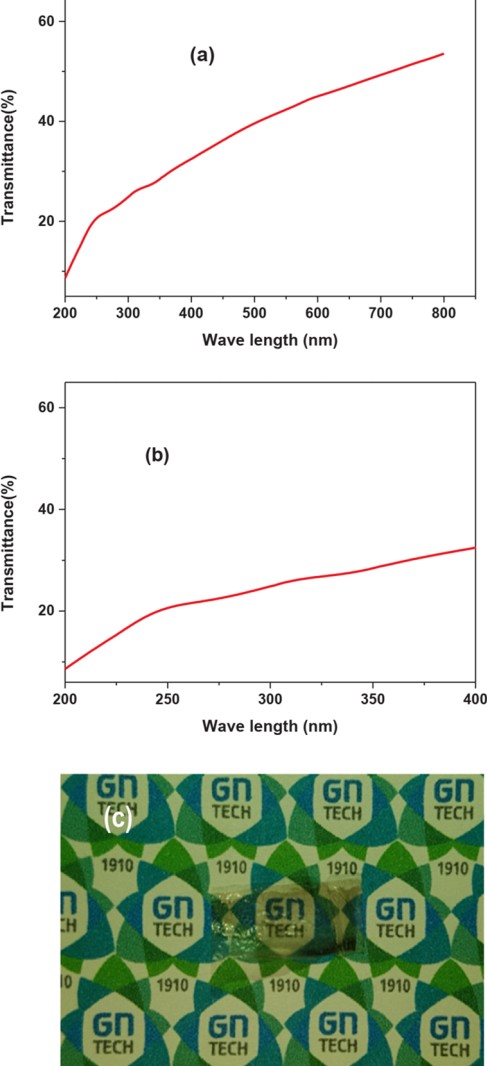

**Figure 4.** (**a**) UV–Vis spectra of the flexible transparent freestanding GLS-GRP-PVA-150 film, (**b**) the UV spectra region of flexible transparent freestanding GLSGRP-PVA-150 film, and (**c**) digital photo of the flexible transparent freestanding GLS-GRP-PVA-150 film reflecting the transparency. Reproduced with permission [29]. Copyright 2021, Elsevier.

Hongtao Zhao et al. found that the pristine polyamide weave fabric has a low capability for blocking UV rays and recorded a UPF value of 2.5 (Table 1). Hence, the coating layer was employed to the fabric surface via the electrophoretic deposition method incorporating the polyamide fabric with graphene oxide (GO) nano-platelets [33]. In this process, polyethyleneimine was utilized to modify the fabric's surface by endowing different polar groups to facilitate the grafting tendency [33]. Afterward, GO was in situ reduced via the green hot-press technique into reduced graphene oxide (rGO). The developed coating layer (polyamide/rGO nanocomposite) afford an excellent UV protection ability with a UPF value of more than 500 (Table 1), which is about 200 times higher than that of the control fabric [33]. Interestingly, durability properties for the treated textile fabrics were attained, even after ten times of laundering [33]. In another study, knit polyester fabrics (PES) were coated with GO, rGO, and eventually rGO/silver nanoparticles via the dip-coating approach. The as-treated fabrics have demonstrated high stretchability, foldability and flexibility. Thus, the polyester fabric coated with nanocomposite of rGO/AgNPs contained little concentration of AgNPs (PES-rGO/Ag) achieved a UV protection ability superior to that of the polyester fabric coated with graphene oxide (PES-GO), and reduced the graphene oxide (PES-rGO) recording reduction in UV transmittance by 73, 43 and 33%, respectively [34]. This enhancement in UV blocking ability could be attributed to the synergistic effect that occurred between rGO and AgNPs. Furthermore, Xiaoning Tang et al. fabricated a multifunctional cotton fabric coating based on GO nanosheets via a vacuum filtration deposition (VFD), and afterward the polyaniline layer was polymerized [35]. The data have revealed that the incorporation of PANI and GO into the cotton fabric (PANI-GO-cotton fabric) affords an outstanding UPF value of 445 with convenient durability, which is superior to that of the GO-based coated cotton fabric and uncoated cotton fabric, as indicated in Table 1 [35]. This displays the role of the PANI layer for blocking UV rays. Additionally, the synergistic UV blocking effect was obtained when the GO sheets were decorated with zinc oxide quantum dot (ZnO QD) and then wrapped with polyvinyl alcohol (PVA) and then utilized as green coating for cotton fabrics [36]. The UPF value of the cotton fabrics coated with GO-ZnO QD-PVA was found to be 61, which is higher than that of the uncoated fabric (15), cotton fabric coated only with GO (40) and cotton fabric coated with ZnO QD alone (34) (Table 1) [36]. In addition to the superior protective ability of the developed textile fabric coating, it displayed a durability feature for 20 consecutive laundering processes [36]. This corroborates that the nanocomposite-based coating affords a superior UV blocking effect compared to the coating with GO alone. Moreover, GO was facilely modified with 2,3-epoxypropyl tri-methyl ammonium chloride (EPTAC) and then reduced by L-Ascorbic acid (L-AA) to provide cotton fabric with the UV protection feature [37]. The UPF values for the uncoated and coated cotton fabrics have been tabulated in Table 1. Interestingly, the UPF values for the cotton fabric coated with rGO was found to be 70. However, for the cationized cotton fabrics coated with reduced rGO, it was detected to be 220. Additionally, the UPF value for synergistically modified cotton fabric-rGO was estimated to be 488 (Table 1) [37]. Additionally, Shu-Dong Wang et al. successfully synthesized a multifunctional silk fabric coating with a good UV protection ability by grafting GO into the fabric surface via hydrogen bonds. The results revealed that the modified silk fabric exhibited more than a 60-fold increment in the UPF value that reached 445 compared to the silk fabric (Table 1) [38]. Another attempt conducted by Bin Zhou et al. found that spraying castor leaves with GO and $TiO_2$ nanoparticles individually on silk textile fabrics enhances the UV protection ability significantly [17]. This is due to the ability of GO and the $TiO_2$ nanoparticles for blocking UV rays by different means. Jia Xu et al. modified GO with dimethyl phosphide and perfluorohexyl iodine chain via the grafting route (GO-multi/cotton) and then coated it on cotton fabrics via the dipping-drying method [39]. Afterward, GO was reduced in situ into rGO inside the matrix of pristine cotton fabric. The results revealed that the UPF values for the treated cotton fabrics of GO-multi/cotton fabric, and rGO/cotton fabric have been significantly improved recording 253 and 500, respectively, compared to 13 for the untreated fabric (Table 1) [39]. This indicates that after treatment, the UV protection

ability was enhanced (50+) for both GO-multi and rGO samples. Furthermore, the study found that the UV protection ability stemmed from the reflection of those destructive rays rather than the absorption [39]. Jeremiah Amesimeku et al. found that uncoated aramid fabric (AF) absorbs UV radiation in the range from 300 to 400 nm, which causes destruction to the bonds holding the fibers together. Therefore, the fabrication of the woven aramid fabric (AF) with the UV resistant coating was executed via doping with polydopamine (PDA)-modified GO with esterification and π-π interaction, followed by the reduction process for the GO to form rGO [40]. The results revealed that the UPF value of the developed fabric (AF-rGO-PDA) was significantly improved, achieving 73 compared to 44 and 37 for AF-PDA and uncoated AF, respectively (Table 1) [40]. Moreover, Feilong Shi et al. have developed a GO-based coating via a silane coupling (crosslinking) agent (KH570) and then coated it on the surface of cotton fabric using a simple dipping–padding–drying method. The developed coating layer affords good reflection to UV rays, recording a UPF value of 187, and this is corroborated with inferior transmittance values in the UVA and UVB of 0.51 and 0.42%, respectively [41]. In another study, Yimin Ji et al. developed a silk coating layer containing GO that further was reduced with L-ascorbic acid (LAA). The UPF value of the developed rGO/silk fabric reached a value of 55 compared to 6 for the uncoated silk fabrics, which is higher than the recommended value by the standard (50+), and this improvement in UV blocking stemmed from the intrinsic features of rGO [42]. Nengyu Pan et al. reported a coating of cotton fabric with GO-modified polymeric N-halamine precursor that was then reduced GO to rGO with L-ascorbic acid (L-AA) that was utilized as a capping agent as well to stabilizing agent for the rGO nanosheets on the coating layer. Afterwards, the coated cotton fabric was chlorinated, and the ability of the coated fabrics for blocking UV rays was significantly increased, recording a UPF value of 132 (Table 1) [43]. Adhering to the same trend, the GO-based coating for silk fabrics was developed and GO was reduced to rGO by sodium hydrosulfite [44]. The data displayed a significant reduction in the transmittance of UVA and UVB and in turn a superior UPF value was attained compared to the uncoated silk fabrics [44]. The enhanced UV protection was attributed to the two-dimensional planar structure of those materials [44]. Pandiyarasan et al. reported a cotton fabric coating layer based on the reduction of GO to rGO into the surface of the cotton fabric via a hydrothermal reduction. Interestingly, the UV protection properties were improved with durable feature recording the UPF values before and after laundering of 443 and 422, respectively [45]. However, Babaahmadi et al. developed a new approach for synthesizing the rGO decorated with tin oxide nanoparticle (rGO/SnO$_2$) nanocomposite and then exploited as UV protective coating layer for the polyethylene terephthalate (PET) fabric [46]. In this approach, SnCl$_2$ was utilized for a dual function as a reducing agent to reduce GO into its reduced form rGO, and as a precursor to synthesize the SnO$_2$ nanoparticles. The UPF for uncoated and coated PET have been calculated based on the transmittance value in the UVA and UVB regions, showing an excellent filtration of those harmful rays even after ten times of laundering. The UPF values have been significantly increased from 34 for pristine PET to 96 for PET/GO and then up to 217 for PET/rGO/SnO$_2$. The significant improvement in the4 UPF value of PET/rGO/SnO$_2$ (Table 1) stemmed from the intrinsic feature of the SnO$_2$ nanoparticles to transmit visible light and filter the UV rays (via basotption) because of their suitable bandgap. Moreover, PET/rGO/SnO$_2$ has exhibited good durability and stability even after being washed ten consecutive times [46]. Furthermore, Mingwei Tian et al. studied the coating of cotton fabric from GO and chitosan via a layer-by-layer electrostatic self-assembly route [16]. In this particular process, the negatively charged GO was considered as a polyanions, and the positively charged chitosan acted as the polycations, and they deposited alternatively on the surface of cotton fabric. The developed coating layer afforded a superior UV protection ability compared to the uncoated fabric and hence, a higher UPF value was attained for the coated fabrics compared to the uncoated one (Table 1) [16]. Interestingly, Amirhosein Berendjchi et al. studied the parameters that affect the capability of the fabrics for shielding UV rays. Thus, they found that the polyethylene terephthalate (PET) fabric basically absorbs UV irradiation depending upon some factors

such as fabric density, construction and inter-fiber spaces, and its UPF value was found to be 23. Consequently, they have coated PET with rGO and then further coated it with a polypyrrole layer (PPY) via in situ polymerization. The choice of PPY was according to the ability of its doped form to absorb UV rays. The dopant narrowed down the band gap and hence facilitated the electronic transition when excited by UV light. The rGO-PPY/PET fabric displayed a UV protection ability with a UPF of 73 [47]. Interestingly, Mingwei Tian et al. prepared a cotton fabric with robust UV protection ability by depositing graphene doped with poly(3,4-ethylenedioxylthiophene): poly (styrenesufonate) (PEDOT: PSS) and chitosan via the layer-by-layer electrostatic self-assembly method [48]. The obtained fabric has revealed a significant UV protection ability with a UPF value of 312, which was up to a 30-fold increment of the pristine fabric (Table 1) [48].

**Table 1.** UPF values for different material-based coated textile fabrics (Based on literature the general uncertainty value of UPF value is $\pm 3\%$. Additionally, equal to or more than 50 is excellent UPF value.

| Sample Code | UPF Value | Ref. |
| --- | --- | --- |
| Polyamide weave fabric | 2.5 | [33] |
| Polyamide weave fabric/rGO | 500 | [33] |
| PANI/GO/cotton fabric | 445 | [35] |
| GO/cotton fabric | 425 | [35] |
| Cotton fabric | 7 | [35] |
| GO/ZnO QD/PVA/cotton fabric | 61 | [36] |
| Uncoated cotton fabric | 15 | [36] |
| Cotton fabric/GO | 40 | [36] |
| Cotton fabric/ZnO QD | 33.6 | [36] |
| Pristine cotton fabric | 24 | [37] |
| Cotton fabric/rGO | 70 | [37] |
| Cationized cotton fabric/rGO | 220 | [37] |
| Synergistically modified cotton fabric/rGO | 488 | [37] |
| Silk fabric/GO | 445 | [38] |
| Silk fabric | 4 | [38] |
| Untreated cotton fabric | 13 | [39] |
| GO-multi/cotton fabric | 253 | [39] |
| rGO/cotton fabric | 500 | [39] |
| AF/rGO/PDA | 73 | [40] |
| AF/PDA | 44 | [40] |
| Pristine AF | 37 | [40] |
| Cotton fabric/GO | 187 | [41] |
| rGO/silk fabric | 55 | [42] |
| Uncoated silk fabric | 6 | [42] |
| rGO/cotton fabric (after chlorination) | 132 | [43] |
| rGO/cotton fabric (before laundering) | 443 | [45] |
| rGO/cotton fabric (after laundering) | 422 | [45] |
| Pristine PET | 34 | [46] |
| PET/GO | 96 | [46] |
| PET/rGO/SnO$_2$ | 217 | [46] |
| Cotton fabric/GO/chitosan | 452 | [16] |
| Uncoated cotton fabric | 9 | [16] |
| PET | 23 | [47] |
| PET/rGO/PPY | 73 | [47] |
| Cotton fabric/graphene/PEDOT:PSS/chitosan | 312 | [48] |
| Pristine cotton fabric | 9 | [48] |
| TiO$_2$ NP-ZnO NP/textile fabric | 58 | [12] |
| TiO$_2$NP-ZnONP | 19 | [12] |
| Uncoated textile fabric | 9 | [12] |
| SiO$_2$NP-AgNPs/cotton fabric | 124 | [10] |
| cotton fabric/binder | 20 | [10] |
| Uncoated cotton fabric | 9 | [10] |

### 2.2. Spherical Nanoparticles-Based Coatings

Spherical nanoparticles such as TiO$_2$ NPs, ZnO NPs and SiO$_2$ NPs have been extensively utilized as UV protective coating for various textile fabrics [49,50]. One of the widely used nanoparticles in fabrics treatments is TiO$_2$ NPs due to its merits such as being antibacterial, self-cleaning and an excellent UV absorber [51,52]. This is due to the suitability of its band gap for absorption of rays in the range from 280–400 nm [12,15,53–55]. Attia et al. reported well dispersed TiO$_2$ NPs- and ZnO NPs-based textile fabrics coating individually and synergistically for various textile fabrics as shown in Figure 5 [12]. The results revealed that the UV protection ability was strongly improved compared to pristine cotton fabrics and coating free of nanoparticles recording UPF values of 58, 9 and 20, respectively. Moreover, outstanding antibacterial properties were attained for the coated textile fabrics [12]. On the other hand, the same group had a further developed textile fabric coating based on SiO$_2$ NPs derived from rice husk biomass and then decorated with silver nanoparticles and exploited as a UV protective layer (Figure 6) [10]. The developed coating affords a strong UPF value for treated cotton fabrics recording 124 compared to 20 for the SiO$_2$ NPs-AgNPs free coating and 9 for the uncoated cotton fabrics [10]. Recently, it was found that the textile fabric texture structure played a significant role on the tendency of the same coating layer for the shielding of harmful UV rays when it utilized SiO$_2$ NPs-chitosan as a sustainable based coating layer, as indicated in Table 2 [2].

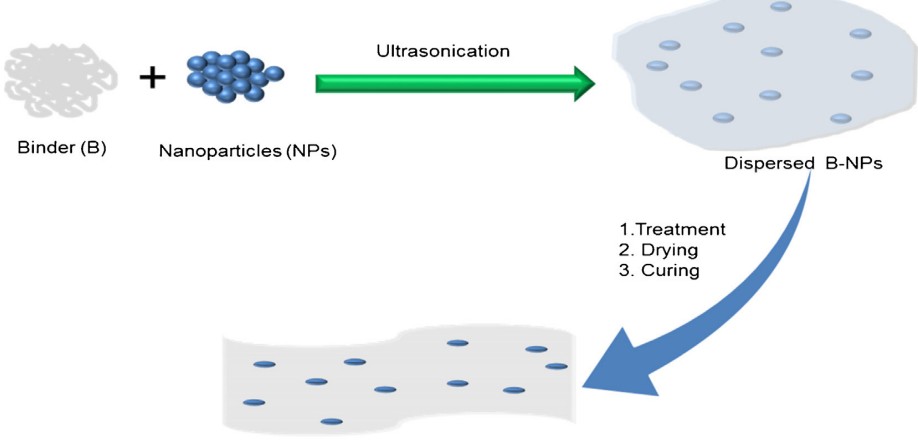

**Figure 5.** Schematic diagram showing the synthesis of coating dispersion and treatment on textile fabrics. Reproduced with permission [12]. Copyright 2017, Elsevier.

On the other hand, the inclusion of the antimicrobial silver nanoparticles (AgNPs) is found to have a UV protection capability against UV rays [10,55]. Therefore, the coating of PET with a nanocomposite based on rGO-AgNPs enhanced the UV protection ability of that textile fabric [55]. Hence, the UPF value for the developed coated fabrics recorded 6145 compared to 34 for pristine cotton [55]. Eventually, the final product demonstrates not only a UV filtering capability, but antibacterial properties as well [55]. Furthermore, a AgNPs-ZnO composite cotton fabric coating was developed using fruit extract and it achieved good UV protection and antibacterial properties [56]. Hence, the UPF value for coated cotton was found to be 70 compared to 7 for the blank cotton sample [56].

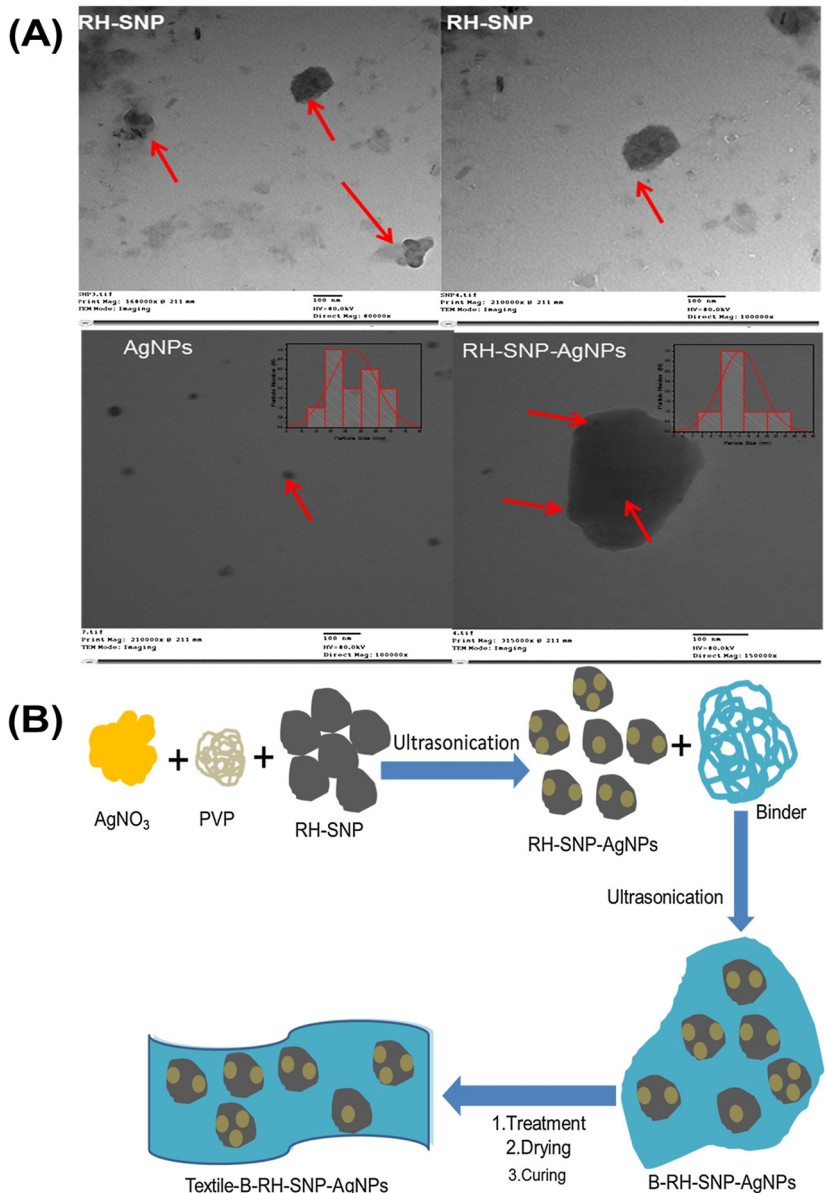

**Figure 6.** (**A**) TEM images of RH-SNP, AgNPs and RH-SNP-AgNPs and insets in (**A**) represent histogram of AgNPs. (**B**) Schematic diagram representing the synthesis of RH-SNP-AgNPs and their coating layer and treatment on textile fabrics. Reproduced with permission [10]. Copyright 2017, Elsevier.

**Table 2.** Ultraviolet protection factor (UPF) values of 280–400 nm of uncoated and coated textile samples. Reproduced with permission [2]. Copyright 2021, Taylor and France.

| Sample Code | UPF Value |
|---|---|
| VW | 3.5 |
| VW-CH-RH-SNP-10 | 5.2 |
| VW-CH-RH-SNP-20 | 6.6 |
| VW-CH-RH-SNP-30 | 12.6 |
| VB | 4.7 |
| VB-CH-RH-SNP-10 | 6 |
| VB-CH-RH-SNP-20 | 6.5 |
| VB-CH-RH-SNP-30 | 5.9 |

**Table 2.** *Cont.*

| Sample Code | UPF Value |
|---|---|
| PSW | 5.7 |
| PSW-CH-RH-SNP-10 | 8.7 |
| PSW-CH-RH-SNP-20 | 13.6 |
| PSW-CH-RH-SNP-30 | 14.7 |
| PSB | 3.6 |
| PSB-CH-RH-SNP-10 | 7.9 |
| PSB-CH-RH-SNP-20 | 10.5 |
| PSB-CH-RH-SNP-30 | 15.4 |

### 2.3. Sustainable Nanotube-Based Coatings

On the other hand, the utilization of a one-dimensional based coating for integrating high UV protection ability to textile fabrics has received attention [57]. Attia et al. developed a textile fabric coating based on naturally abundant halloysite nanotubes (HNTs) decorated with molokhia extract and then coated on linen textile fabrics, which are mainly used in historical textile conservation (Figure 7) [58]. The green coating was uniformly dispersed on the linen fabric surface and bonded with the linen surface via supramolecular interactions. Hence, the developed linen fabrics achieved significant protection against harmful UV rays, recording an enhancement in UPF factor by 57% compared to the uncoated fabrics [58].

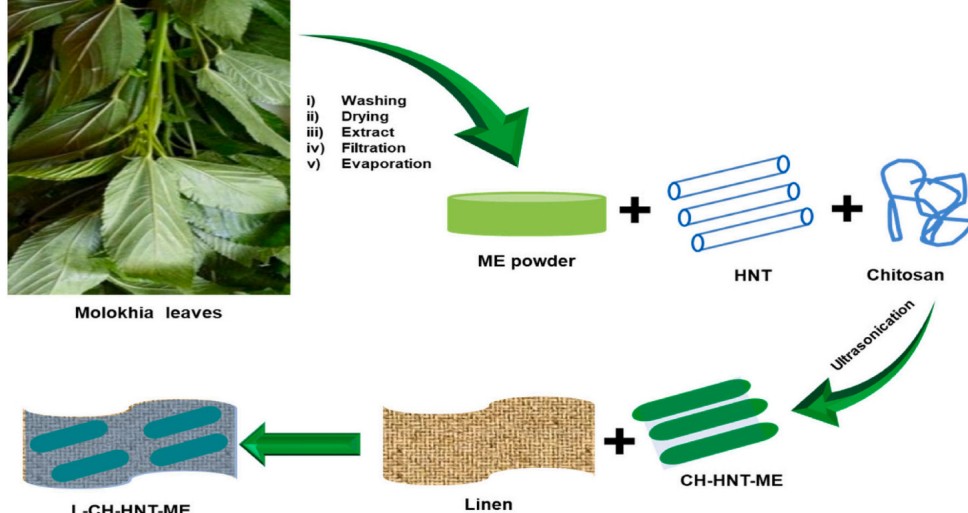

**Figure 7.** Schematic diagram representing the synthesis of green textile coating CH-HNT-ME and their linen textile fabric composite L-CH-HNT-ME. Reproduced with permission [58]. Copyright 2022, Elsevier.

Interestingly, our group recently developed a smart and innovative transparent nanocomposite layer composed from cellulose nanocrystals derived from natural cotton fibers and wrapped on HNTs via the green and facile approach, as displayed in Figure 8 [59]. Then, the transparent green layer was sprayed over historical written paper (HP) and it integrated a UV protection ability superior to HP, recording a UPF value of 58 compared to 39 for pristine HP without altering the visual properties of the written words [60]. This approach adhered to sustainable development goals (SDGs) that encourage dependence on renewable precursors [60].

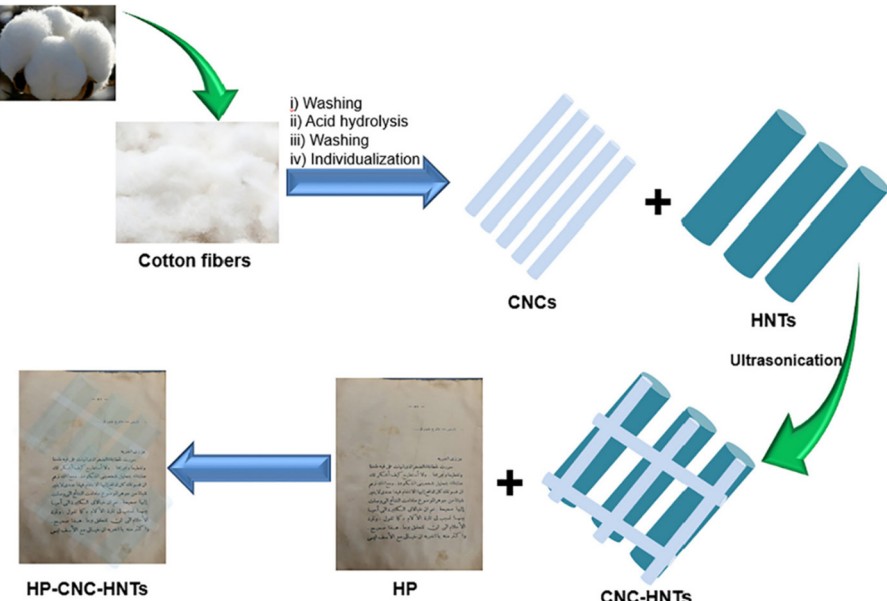

**Figure 8.** Schematic diagram representing the synthesis of CNC-HNTs conservation layer and their usage in consolidation to historical paper. Reproduced with permission [59]. Copyright 2022, Elsevier.

## 3. Mechanistic UV Protection Action

The UV protection always depends on absorption of UV rays by UV protective agents in the coating layer and/or reflection of UV rays. Thus, for materials following the absorption trend, the tendency of absorption is mainly governed by the gap between the valence and the conduction bands of these materials [12,54]. However, for those kinds of materials that primarily work based on the reflection trend, the structure and crystallinity played a key role for the performance of their UV ray reflection [29]. Therefore, to achieve good UV blocking performance for the first class of UV protective agents, the band gap of semiconducting materials has to be tuned via engineering of the band gap. This is usually carried out by decorating and doping of such kind of materials with peculiar metal nanoparticles bridging the gap and in turn narrowing down the band gap to be suitable for the absorption of UV rays [16,49,50,54,61,62]. On the other hand, for UV reflection action materials, the fabrication of nanocomposites-based coating layer rather than only one component provides a new synergistic feature for the reflection of harmful UV rays [29,34–37]. In conclusion, the UV blocking actions by the various sustainable textile fabric coatings are summarized in Figure 9.

Recently, wool grease was extracted from coarse wool fleece using conventional heating or microwave irradiation. Afterward, it was utilized at different concentrations of lanoline as a binder for the pigment printing of wool, polyester fabric, polyester/wool (65/35), and polyester/cotton (65/35) using the flat screen technique. Interestingly, these printed fabrics record a significant enhancement in UPF values compared to their original fabrics [63].

Interestingly, the application of those functionalized textiles in electronics can be found through the utilization of those coated fabrics with UV protective layers in different organic electronic devices such as organic solar cells (OSCs). Such textiles can be utilized as both substrates and encapsulants to protect those solar cells used for outdoor applications. Therefore, an optimized fabrication approach was used to entirely spray coating technique to fabricate the OSCs with a power conversion efficiency of 0.4%. Initially, interface layer is deposited on the woven textile to form a smooth supporting layer for the subsequent spray-coated functional layers. Then, an encapsulant was deposited on the top of the fabricated OSCs to enhance the durability and lifetime of the as-fabricated OSCs [64]. On the other hand, metal–organic decomposition (MOD) ink is another promising and alternative technique for utilization in coating of textiles with metal for electronic applications [65].

This process implies dissolving of the metal source (eg. silver) in a convenient solvent and then the evaporation of solvent [66,67]. The choice of solvent is very critical, because after the evaporation of the solvent and the decomposition of the organic complexes the volume of MOD ink is significantly diminished [68]. This approach is highly recommended for flexible printed electronic applications [69].

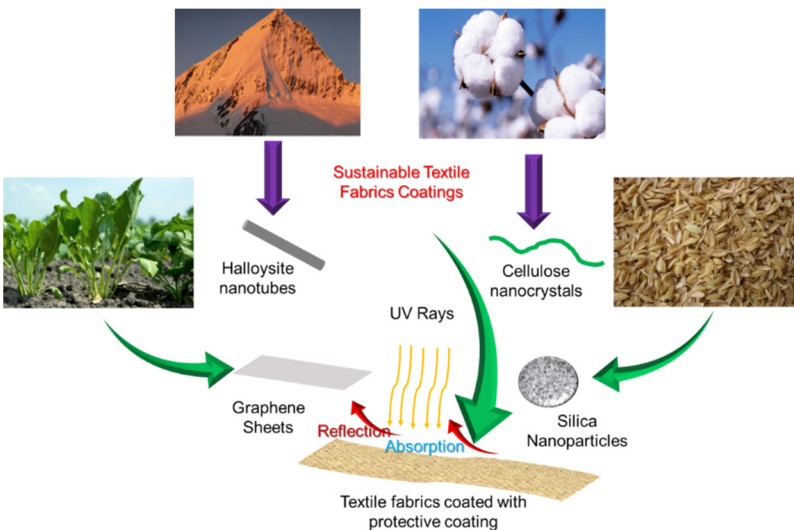

**Figure 9.** Schematic diagram representing the routes for sustainable textile fabric coatings and their UV rays blocking approaches.

## 4. Conclusions

Harmful UV rays play a significant role in shorting the lifetime and efficiency of various natural and synthetic materials. Therefore, filtering them is a mandatory step before commercialization of these materials. Among these sensitive materials are textile fabrics (natural and synthetic) that are involved in various applications, including the preservation of the history of a nation. In this review, recent developments relating to sustainable textile fabric coatings for the protection of various textile fabrics were discussed. Some focus was devoted to the exploitation of graphene sheet-based coatings for UV ray protection as a unique 2D material derived from natural precursors. Moreover, the potential of naturally abundant inorganic nanotubes, silica nanoparticles and cellulose nanocrystals for filtrating harmful UV rays and their negative effect was studied. Additionally, the main UV protection actions were proposed.

**Author Contributions:** Conceptualization, N.F.A. and S.E.A.E.; methodology, N.F.A. and R.O.; software, A.K.; validation, A.G.A.-S. and H.A.; investigation, N.F.A.; resources, A.K.; data curation, S.E.A.E.; writing—original draft preparation, N.F.A., S.E.A.E. and R.O.; writing—review and editing, N.F.A.; visualization, N.F.A. and R.O.; supervision, N.F.A.; project administration, A.K.; funding acquisition, A.G.A.-S. and H.A. All authors have read and agreed to the published version of the manuscript.

**Funding:** The authors acknowledge support and funding of King Khalid University through the Research Center for Advanced Materials Science (RCAMS) under grant no: RCAMS/KKU/0010/21.

**Institutional Review Board Statement:** Not applicable.

**Informed Consent Statement:** Not applicable.

**Data Availability Statement:** Not applicable.

**Conflicts of Interest:** The authors declare no conflict of interest.

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
