# Peer review of "Recent Advances of Sustainable Textile Fabric Coatings for UV Protection Properties"

_coatings, doi:10.3390/coatings12101597_

Round 1
Reviewer 1 Report
In this work, the authors reviewed sustainable coatings for textile fabrics, the synthesis routes, and the merits concerning ultraviolet ray protection. Overall, the manuscript is informative and thus deserves publication in Coatings. However, the quality of the presentation and the English language must be improved before its publication. Below are some comments for the authors’ consideration.
Major comments
1. Section 1 lacks an introduction to sustainable textile fabric coatings. Please considering add a paragraph for this.
2. The first paragraph of Section 2 should mention all coatings to be reviewed in this section.
3. The paragraph spanning lines 106--218 is hard to digest due to a lack of organization. Is it feasible to tabulate the information therein? It is also helpful to have more comparisons between different studies in the text.
4. Section 3 is not informative enough. Can you elaborate more on this section?
5. In lines 45 and 46, do you mean oxygen and ozone molecules by “oxygen atoms”?
Minor comments
6. A verb is missing in the sentence “Among these functions …” in lines 19 and 20.
7. The sentence “There are various textile …” in lines 71—73 is hard to follow. Please rewrite it to be concise.
8. In line 74, the word “Thus” is misused. There is no clear causal relationship between the sentences before and after it.
9. In lines 80—81, the phrase “it is demand” is problematic.
10. In lines 89—90, the sentence “Thus, displaying …” is very confusing.
11. In line 102, a right parenthesis symbol “)” is missing for “(a”.
12. In the caption of Figure 5, it is better to move the figure labels “(a)”, “(b)”, and “(c)” to the beginning of the corresponding explanatory sentences.
13. In Section 2.2, a space needs to be added between “TiO2” and “NPs” in “TiO2NPs”. Similarly for “ZnONPs” and “SiO2NPs”.
14. In line 223, the meaning of the word “This” is ambiguous.
15. The newline character at the end of line 224 needs to be removed.
16. In line 233, “SiO” should be “SiO2”.
Author Response
Responses to the Comments of Reviewer #1
Reviewer #1: In this work, the authors reviewed sustainable coatings for textile fabrics, the synthesis routes, and the merits concerning ultraviolet ray protection. Overall, the manuscript is informative and thus deserves publication in Coatings. However, the quality of the presentation and the English language must be improved before its publication. Below are some comments for the authors’ consideration.
We thank the reviewer for imparting his valuable time in reviewing the manuscript and
acknowledging the importance of the work and his useful comments and recommendation for publication.
Major comments
- Section 1 lacks an introduction to sustainable textile fabric coatings. Please considering add a paragraph for this.
(Authors’ Response) Thank you for reviewer comment. This has been considered in the revised version and highlighted. Thus, new section and new reference were inserted and highlighted. Therefore, the following section was added as below.
Inserted Section
(Page 2) Recently, fabrication of sustainable textile fabrics coatings mainly derived from renewable precursors has got strong attentions from researchers to afford protection to textiles and human beings from serious effects of harmful UV rays [10]. This trend, adhered to sustainable development goals (SDGs) which primarily implies the valorization of renewable and abundance biomass wastes to high-tech applications affording sustainability and cost-effectiveness [12].
Inserted Reference
- United Nations Transforming Our World: The 2030 Agenda for sustainable Development. Available online: https://sdgs.un.org/ 2030agenda (accessed on 12 Oct 2022)
- The first paragraph of Section 2 should mention all coatings to be reviewed in this section.
(Authors’ Response) Thank you for reviewer comment. This has been considered in the revised version and highlighted. Thus, new section has been inserted and highlighted. Therefore, the following section was added as below.
Inserted Section
(Page 2) Therefore, several graphene, spherical nanoparticles, sustainable inorganic nanotubes textile fabrics-based coatings implemented using various routes such as immersion, electrophoretic deposition, vacuum filtration deposition, dip-coating and layer-by-layer techniques will be studied.
- The paragraph spanning lines 106--218 is hard to digest due to a lack of organization. Is it feasible to tabulate the information therein? It is also helpful to have more comparisons between different studies in the text.
(Authors’ Response) Thank you for reviewer comment. This has been considered in the revised version and highlighted. Therefore, significant enrichment in discussion was carried out for data and highlighted in revised version. Additionally, the UPF values of reported compared samples were tabulated in Table 1 for clarification and highlighted.
Inserted Table 1.
Table 1. UPF values for different materials based coated textile fabrics
|
Sample Code |
UPF Value |
Ref |
|
Polyamide weave fabric |
2.5 |
29 |
|
Polyamide weave fabric/ rGO |
> 500 |
29 |
|
PANI/ GO/ cotton fabric |
445 |
31 |
|
GO/ cotton fabric |
425 |
31 |
|
Cotton fabric |
7 |
31 |
|
GO/ ZnO QD/ PVA/ cotton fabric |
61 |
32 |
|
Uncoated cotton fabric |
15 |
32 |
|
Cotton fabric/ GO |
40 |
32 |
|
Cotton fabric/ ZnO QD |
33.6 |
32 |
|
Pristine cotton fabric |
24 |
33 |
|
Cotton fabric/ rGO |
70 |
33 |
|
Cationized cotton fabric/ rGO |
220 |
33 |
|
Synergistically modified cotton fabric/ rGO |
488 |
33 |
|
Silk fabric/ GO |
445 |
34 |
|
Silk fabric |
4 |
34 |
|
Untreated cotton fabric |
13 |
36 |
|
GO-multi/ cotton fabric |
253 |
36 |
|
rGO/ cotton fabric |
500 |
36 |
|
AF/ rGO/ PDA |
73 |
37 |
|
AF/ PDA |
44 |
37 |
|
Pristine AF |
37 |
37 |
|
Cotton fabric/ GO |
187 |
38 |
|
rGO/ silk fabric |
55 |
39 |
|
Uncoated silk fabric |
6 |
39 |
|
rGO/ cotton fabric (after chlorination) |
132 |
40 |
|
rGO/ cotton fabric (before laundering) |
443 |
42 |
|
rGO/ cotton fabric (after laundering) |
422 |
42 |
|
Pristine PET |
34 |
43 |
|
PET/ GO |
96 |
43 |
|
PET/ rGO/ SnO2 |
217 |
43 |
|
Cotton fabric/ GO/ chitosan |
452 |
44 |
|
Uncoated cotton fabric |
9 |
44 |
|
PET |
23 |
45 |
|
PET/ rGO/ PPY |
73 |
45 |
|
Cotton fabric/ graphene/ PEDOT:PSS/ chitosan |
312 |
46 |
|
Pristine cotton fabric |
9 |
46 |
|
TiO2 NP-ZnO NP/ textile fabric |
58 |
13 |
|
TiO2NP-ZnONP |
19 |
13 |
|
Uncoated textile fabric |
9 |
13 |
|
SnO2NP-AgNPs/ cotton fabric |
124 |
10 |
|
cotton fabric/binder |
20 |
10 |
|
Uncoated cotton fabric |
9 |
10 |
|
HP-cellulose nanocrystal/ HNTs |
58 |
61 |
|
Pristine HP |
39 |
61 |
- Section 3 is not informative enough. Can you elaborate more on this section?
(Authors’ Response) Thank you for reviewer comment. This has been considered in the revised version and highlighted. Hence, new section, figure and references were inserted and highlighted in revised version. Therefore, the following section, references and figure were added as below.
Inserted Section
(Page 12) Therefore, to achieve good UV blocking performance for the first class of UV protective agents, the band gap of semiconducting materials has to be tune via engineering of band gap. This is usually carried out by decorating and doping such kind of materials with peculiar metal nanoparticles bridging the gap and in turn narrow down the band gap to be suitable for absorption of UV rays [46-49,63-65]. On the other hand, for UV reflection action materials, the fabrication of nanocomposites-based coating layer ra-ther than only one component providing new synergistic feature for reflection of harmful UV rays [27,33-36]. In conclusion the UV blocking actions by various sustain-able textile fabric coatings is summarized in Fig.9.
Inserted References
- El-Sayed, W. G., Attia, N. F., Ismail, I., El-Khayat, M., Nogami, M., Abdel-Mottaleb. M. S. A.
Innovative and cost-effective nanodiamond based molten salt nanocomposite as efficient heat transfer fluid and thermal energy storage media. Renew. Energy 2021, 177, 596-602.
- Radetić, M. Functionalization of textile materials with TiO2 nanoparticles. J. Photochem.
Photob. C: Photochem. Rev.2013,16,62-76.
Inserted Figure
Figure 9. Schematic diagram representing the routes for sustainable textile fabric coatings and their UV rays blocking approaches.
- In lines 45 and 46, do you mean oxygen and ozone molecules by “oxygen atoms”?
(Authors’ Response) Thank you for reviewer comment. Yes, we mean by oxygen molecules oxygen and ozone. X
Minor comments
- A verb is missing in the sentence “Among these functions …” in lines 19 and 20.
(Authors’ Response) Thank you for reviewer comment. This has been considered in revised version and highlighted in abstract section.
- The sentence “There are various textile …” in lines 71—73 is hard to follow. Please rewrite it to be concise.
(Authors’ Response) Thank you for reviewer comment. This has been considered in revised version and highlighted (Page 2).
- In line 74, the word “Thus” is misused. There is no clear causal relationship between the sentences before and after it.
(Authors’ Response) Thank you for reviewer comment. This has been considered in revised version and highlighted .
- In lines 80—81, the phrase “it is demand” is problematic.
(Authors’ Response) Thank you for reviewer comment. This has been considered in revised version and highlighted .
- In lines 89—90, the sentence “Thus, displaying …” is very confusing.
(Authors’ Response) Thank you for reviewer comment. This has been considered in revised version and this sentence was rewritten and highlighted .
- In line 102, a right parenthesis symbol “)” is missing for “(a”.
(Authors’ Response) Thank you for reviewer comment. This has been considered in revised version and highlighted.
- In the caption of Figure 5, it is better to move the figure labels “(a)”, “(b)”, and “(c)” to the beginning of the corresponding explanatory sentences.
(Authors’ Response) Thank you for reviewer comment. This has been considered in revised version and highlighted.
- In Section 2.2, a space needs to be added between “TiO2” and “NPs” in “TiO2NPs”. Similarly for “ZnONPs” and “SiO2NPs”.
(Authors’ Response) Thank you for reviewer comment. This has been considered in revised version and highlighted.
- In line 223, the meaning of the word “This” is ambiguous.
(Authors’ Response) Thank you for reviewer comment. This has been considered in revised version and highlighted.
- The newline character at the end of line 224 needs to be removed.
(Authors’ Response) Thank you for reviewer comment. This has been considered in revised version and highlighted.
- In line 233, “SiO” should be “SiO2”.
(Authors’ Response) Thank you for reviewer comment. This has been considered in revised version and highlighted.

Reviewer 2 Report
1) presentation-related comments
In Chapters 2.1 and 2.2, the authors compile findings on the topic in narrative text. As a reader, I would expect that the findings are summarized in new tables
Technique (short description) // Ref. // UPF value +/- uncertainty // Optional: Application
Please do so as this will be the original value upgrade of your article
2) Topic related comments:
Please check the following questions and
- Does inclusion of carbon nanofibers into yarns have any protection effect (possibly in other spectral ranges?)
- Does inclusion of (antimicrobial) silver into yarns have any protection effect?
- Which approaches of functional printing (e.g. inkjet printing for patterned UV coating) are applied in the field?
- Which links does the topic have to the field of "woven electronics" (UV protection and elementary functionalities at the same time)?
3) line-by-line comments
line 16: check e-mail domain "yahooo", prefer institutional e-mail address
lines 31-69: the Introduction should contain a chart on the various UV ranges discussed in the text
line 66 "feature article" vs line 74: "review" please decide for one format ("Review article" according to headline)
line 72: the term "harmful" should be explained (to whom? in what respect?)
line 74: a verb is missing in this sentence, may be omitted, or intro may regard the chapter only
line 81: top-down and bottom-up (no capitalization required)
line 92: remove typo "."
lines 101 ff: Fig. 5
panels (A) and (B): correct labels
line 229: 3x UPF please specify uncertainty values (regards also other occurrences in the paper, your usage of different No. digits at different occurrences inclines variable precision)
lines 240 ff: Fig. 6
Aspect ratio of figure seems to be distorted. Please check and put in correct original aspect ratio.
Transmittance (%) vs. Wavelength (nm)
panel (C) include scalebar (or write in text)
line 245 Table 1:
Explain UPF in the table headline
Reproduce the table in table format, not as graphics
Author Response
Responses to the Comments of Reviewer #2
Reviewer #2: 1) presentation-related comments
We thank the reviewer for imparting his valuable time in reviewing the manuscript
and his useful comments and recommendation for publication.
In Chapters 2.1 and 2.2, the authors compile findings on the topic in narrative text. As a reader, I would expect that the findings are summarized in new tables
(Authors’ Response) Thank you for reviewer comment. This has been considered in the revised version and highlighted. Therefore, significant enrichment in discussion was carried out for data and highlighted in revised version. Additionally, the UPF values of reported compared samples were tabulated in Table 1 for clarification and highlighted.
Inserted Table 1
Table 1. UPF values for different materials based coated textile fabrics
|
Sample Code |
UPF Value |
Ref |
|
Polyamide weave fabric |
2.5 |
29 |
|
Polyamide weave fabric/ rGO |
> 500 |
29 |
|
PANI/ GO/ cotton fabric |
445 |
31 |
|
GO/ cotton fabric |
425 |
31 |
|
Cotton fabric |
7 |
31 |
|
GO/ ZnO QD/ PVA/ cotton fabric |
61 |
32 |
|
Uncoated cotton fabric |
15 |
32 |
|
Cotton fabric/ GO |
40 |
32 |
|
Cotton fabric/ ZnO QD |
33.6 |
32 |
|
Pristine cotton fabric |
24 |
33 |
|
Cotton fabric/ rGO |
70 |
33 |
|
Cationized cotton fabric/ rGO |
220 |
33 |
|
Synergistically modified cotton fabric/ rGO |
488 |
33 |
|
Silk fabric/ GO |
445 |
34 |
|
Silk fabric |
4 |
34 |
|
Untreated cotton fabric |
13 |
36 |
|
GO-multi/ cotton fabric |
253 |
36 |
|
rGO/ cotton fabric |
500 |
36 |
|
AF/ rGO/ PDA |
73 |
37 |
|
AF/ PDA |
44 |
37 |
|
Pristine AF |
37 |
37 |
|
Cotton fabric/ GO |
187 |
38 |
|
rGO/ silk fabric |
55 |
39 |
|
Uncoated silk fabric |
6 |
39 |
|
rGO/ cotton fabric (after chlorination) |
132 |
40 |
|
rGO/ cotton fabric (before laundering) |
443 |
42 |
|
rGO/ cotton fabric (after laundering) |
422 |
42 |
|
Pristine PET |
34 |
43 |
|
PET/ GO |
96 |
43 |
|
PET/ rGO/ SnO2 |
217 |
43 |
|
Cotton fabric/ GO/ chitosan |
452 |
44 |
|
Uncoated cotton fabric |
9 |
44 |
|
PET |
23 |
45 |
|
PET/ rGO/ PPY |
73 |
45 |
|
Cotton fabric/ graphene/ PEDOT:PSS/ chitosan |
312 |
46 |
|
Pristine cotton fabric |
9 |
46 |
|
TiO2 NP-ZnO NP/ textile fabric |
58 |
13 |
|
TiO2NP-ZnONP |
19 |
13 |
|
Uncoated textile fabric |
9 |
13 |
|
SnO2NP-AgNPs/ cotton fabric |
124 |
10 |
|
cotton fabric/binder |
20 |
10 |
|
Uncoated cotton fabric |
9 |
10 |
|
HP-cellulose nanocrystal/ HNTs |
58 |
61 |
|
Pristine HP |
39 |
61 |
Technique (short description) // Ref. // UPF value +/- uncertainty // Optional: Application
Please do so as this will be the original value upgrade of your article
(Authors’ Response) Thank you for reviewer comment. This has been considered in the revised version and highlighted. Thus, new section and references has been inserted and highlighted. Therefore, the following section was added as below.
Inserted Section
(Page 2) Interestingly, UPF value can be straightforwardly evaluated in the range of 280 nm up to 400 nm based on Australian/ New Zealand Standard (AS/ NZS 4399:1996) according to the following equation:
UPF =
Where Eλ refers to the relative erythemal spectral effectiveness (the action spec-trum), Sλ is the solar UV spectral irradiance for a typical summer period during midday in a clear sky condition, dλ indicates the increment in the wavelength, λ represents the wavelength (nm), and Tλ is the spectral transmittance of the specimen determined by the UV/ Vis spectrophotometer equipment [16-17]. The uncertainty of measurement is mainly depending on the accuracy and pression of spectrophotometer used, therefore vary from one to another.
Inserted References
- Babaahmadi, V., Majid, M. Reduced graphene oxide/SnO2 nanocomposite on PET surface:
Synthesis,characterization and application as an electro-conductive and ultraviolet blocking textile. Colloids Surf. A: Physicochem. Eng. Aspect 2016, 506, 507-513.
- Zhou, B., Wang, H., Zhou, H., Wang, K., Wang, S. Natural flat cocoon materials constructed by erisilkworm with high strength and excellent anti-ultraviolet performance. J. Eng. Fibers Fabrics 2020, 15, 1558925020978652.
2) Topic related comments:
Please check the following questions and
- Does inclusion of carbon nanofibers into yarns have any protection effect (possibly in other spectral ranges?)
(Authors’ Response) Thank you for reviewer comment and bring our attention to this point. Actually, there is no report so far studied the inclusion of carbon nanofibers in textile fabric coating for UV blocking. However, this is could be interesting point to study and carbon nanofibers could have sp3/sp2 hybridization and could influence and UV protection and achieve positive results. Also, precursors of carbon nanofibers will play a role. Therefore, our future study will be dedicated for this point.
- Does inclusion of (antimicrobial) silver into yarns have any protection effect?
(Authors’ Response) Thank you for reviewer comment and bring our attention to this point. Yes, the inclusion of AgNPs into yarns displayed good UV protection and antibacterial properties as well. Hence, new section and references are inserted in revised version enriched this point. Therefore, the following parts were added.
Inserted Section.
(Page 10) On the other hand, it is found that the inclusion of the antimicrobial silver nanoparticles (AgNPs) has UV protection capability against UV rays [57]. Therefore, the coating of PET with nanocomposite based on rGO-AgNPs enhanced the UV protection ability of that textile fabric [57]. Hence, the UPF value for the developed coated fabrics record 6145 compared to 34 for pristine cotton [57]. Eventually, the final product demonstrates not only a UV filtering capability, but antibacterial properties as well [57]. Over and above, AgNPs-ZnO composite cotton fabric coating was developed using fruit extract achieving good UV protection and antibacterial properties [58]. Hence, the UPF for coated cotton was found to be 70 compare to 7 for blank cotton sample [58].
Inserted References.
- Babaahmadi, V., Abuzade, R. A., Majid, M. Enhanced ultraviolet‐protective textiles based on
reduced graphene oxide‐silver nanocomposites on polyethylene terephthalate using ultrasonic‐assisted in‐situ thermal synthesis. J. Appl. Polym. Sci. 2022, 139, 52196.
- Porrawatkul, P., Pimsen, R., Kuyyogsuy, A., Teppaya, N., Noypha,A., Chanthai,
S.,Nuengmatcha, P. Micro-wave-assisted synthesis of Ag/ZnO nanoparticles using Averrhoa carambola fruit extract as the reducing agent and their ap-plication in cotton fabrics with antibacterial and UV-protection properties. RSC Adv. 2022,12, 15008-15019.
- Which approaches of functional printing (e.g. inkjet printing for patterned UV coating) are applied in the field?
(Authors’ Response) Thank you for reviewer comment. Actually, the conventional heating or microwave irradiation can be utilized for that purpose. Recently, wool grease was extracted from coarse wool fleece using conventional heating or microwave irradiation. Afterward, has utilized at different concentrations of lanoline as a binder for pigment printing of wool, polyester fabric, polyester/ wool (65/ 35), and polyester/ cotton (65/ 35) using flat screen technique. Interestingly, these printed fabrics record a significant enhancement in UPF values compared to their original fabrics [66]. Hence, the following section with references were inserted in revised version and highlighted.
Inserted Section
(Page 13) Recently, wool grease was extracted from coarse wool fleece using conventional heat-ing or microwave irradiation. Afterward, it has utilized at different concentrations of lanoline as a binder for pigment printing of wool, polyester fabric, polyester/ wool (65/ 35), and polyester/ cotton (65/ 35) using flat screen technique. Interestingly, these printed fabrics record a significant enhancement in UPF values compared to their original fabrics [66].
Inserted Reference
- El-Shemy, N. S., El-Hawary, N. S., Haggag, K., ElSayed, H. Utilization of lanolin in microwave-assisted pigment printing of textiles. Egyptian J. Chem. 2020, 63, 3259-3269.
- Which links does the topic have to the field of "woven electronics" (UV protection and elementary functionalities at the same time)?
(Authors’ Response) Thank you for reviewer comment and bring our attention to this point. The link between those functionalized textiles and electronics can be found through the utilization of those coated fabrics with UV protective layers in different organic electronic devices like organic solar cells (OSCs). Such textiles can be utilized as both substrates and encapsulants to protect those solar cells used for outdoor applications. Therefore, an optimized fabrication approach was used to entirely spray coating technique to fabricate OSCs with a power conversion efficiency of 0.4%. Initially, interface layer is deposited on the woven textile to form a smooth supporting layer for the subsequent spray-coated functional layers. Then, an encapsulant has been deposited on the top of the fabricated OSCs which enhances the durability and life time of the as-fabricated OSCs [67]. Hence, the following section with references were inserted in revised version and highlighted.
Inserted Section
(Page 14) Interestingly, the application of those functionalized textiles in electronics can be found through the utilization of those coated fabrics with UV protective layers in dif-ferent organic electronic devices like organic solar cells (OSCs). Such textiles can be utilized as both substrates and encapsulants to protect those solar cells used for out-door applications. Therefore, an optimized fabrication approach was used to entirely spray coating technique to fabricate OSCs with a power conversion efficiency of 0.4%. Initially, interface layer is deposited on the woven textile to form a smooth supporting layer for the subsequent spray-coated functional layers. Then, an encapsulant has been deposited on the top of the fabricated OSCs which enhances the durability and life time of the as-fabricated OSCs [67].
Inserted Reference
67 Li, Y., Arumugam, S., , Krishnan, C., Charlton, M D.B., Beeby, S. P. Encapsulated textile organic solar cells fabricated by spray coating. ChemistrySelect 2019, 4, 407-412.
3) line-by-line comments
line 16: check e-mail domain "yahooo", prefer institutional e-mail address
(Authors’ Response) Thank you for reviewer comment. This has been considered and institutional e-mail address was inserted in revised version and highlighted.
lines 31-69: the Introduction should contain a chart on the various UV ranges discussed in the text
(Authors’ Response) Thank you for reviewer comment. This has been considered in revised version and introduction section was enriched displaying the UV range focusing on it and how to measure and so on. Therefore, the following sections were inserted and highlighted in revised version.
Inserted Section
(Page 2) Interestingly, the UPF values mainly measured in the rage of 280-400 nm [8].
(Page 2) Interestingly, UPF value can be straightforwardly evaluated in the range of 280 nm up to 400 nm based on Australian/ New Zealand Standard (AS/ NZS 4399:1996) according to the following equation:
UPF =
Where Eλ refers to the relative erythemal spectral effectiveness (the action spec-trum), Sλ is the solar UV spectral irradiance for a typical summer period during midday in a clear sky condition, dλ indicates the increment in the wavelength, λ represents the wavelength (nm), and Tλ is the spectral transmittance of the specimen determined by the UV/ Vis spectrophotometer equipment [16-17]. The uncertainty of measurement is mainly depending on the accuracy and pression of spectrophotometer used, therefore vary from one to another.
Inserted References
- Babaahmadi, V., Majid, M. Reduced graphene oxide/SnO2 nanocomposite on PET surface:
Synthesis,characterization and application as an electro-conductive and ultraviolet blocking textile. Colloids Surf. A: Physicochem. Eng. Aspect 2016, 506, 507-513.
- Zhou, B., Wang, H., Zhou, H., Wang, K., Wang, S. Natural flat cocoon materials constructed by erisilkworm with high strength and excellent anti-ultraviolet performance. J. Eng. Fibers Fabrics 2020, 15, 1558925020978652.
line 66 "feature article" vs line 74: "review" please decide for one format ("Review article" according to headline)
(Authors’ Response) Thank you for reviewer comment. This has been considered in revised version and review word was used.
line 72: the term "harmful" should be explained (to whom? in what respect?)
(Authors’ Response) Thank you for reviewer comment. Actually, the UV rays are harmful to human and other living things and this clearly explained in introduction section as highlighted below.
(Page 2) The UV radiation affects humans and other living things negatively by inducing different physiological impacts with a plethora of acute and slow-rated consequences depending on the wavelength and energy [7]. UV-A is less energetic than UV-B; however, it can penetrate deeper through the second layer of the skin and the dermis, due to its longer wavelength [4-5] and hence causes the generation of reactive oxygen species (ROS) [7] and reactive nitrogen species (RNS) which in turn alter the lipids, proteins, and DNA of the cells [5-6]. These oxidative damages attributed by those highly reactive molecules and may result in wrinkle formation and cause aging of the skin, and may cause the development of skin cancer as a result of immunosuppression against infections [7] especially in the range from 360 nm to 380 nm [4]. On the other hand, UV-A exhibits lesser mutation effects than UV-B [5]
line 74: a verb is missing in this sentence, may be omitted, or intro may regard the chapter only
(Authors’ Response) Thank you for reviewer comment. This has been considered in revised version and highlighted .
line 81: top-down and bottom-up (no capitalization required)
(Authors’ Response) Thank you for reviewer comment. This has been considered in revised version and highlighted .
line 92: remove typo "."
(Authors’ Response) Thank you for reviewer comment. This has been considered in revised version and highlighted .
lines 101 ff: Fig. 5 panels (A) and (B): correct labels
(Authors’ Response) Thank you for reviewer comment. This has been considered in revised version and highlighted .
line 229: 3x UPF please specify uncertainty values (regards also other occurrences in the paper, your usage of different No. digits at different occurrences inclines variable precision)
(Authors’ Response) Thank you for reviewer comment. This has been considered in revised version and highlighted and all values were rounded using same procedure.
lines 240 ff: Fig. 6 Aspect ratio of figure seems to be distorted. Please check and put in correct original aspect ratio.
(Authors’ Response) Thank you for reviewer comment. This has been considered in revised version and highlighted.
line 245 Table 1:
Explain UPF in the table headline
Reproduce the table in table format, not as graphics
(Authors’ Response) Thank you for reviewer comment. This has been considered in revised version and UPF was explained in Table 2 headline and Table 2 was reproduced again and highlighted.
Table 2. Ultraviolet protection factor (UPF) values of uncoated and coated textile samples. Reproduced with permission [2]. Copyright 2021, Taylor and France
|
Sample Code |
UPF Value |
|
VW |
3.5 |
|
VW-CH-RH-SNP-10 |
5.2 |
|
VW-CH-RH-SNP-20 |
6.6 |
|
VW-CH-RH-SNP-30 |
12.6 |
|
VB |
4.7 |
|
VB-CH-RH-SNP-10 |
6 |
|
VB-CH-RH-SNP-20 |
6.5 |
|
VB-CH-RH-SNP-30 |
5.9 |
|
PSW |
5.7 |
|
PSW-CH-RH-SNP-10 |
8.7 |
|
PSW-CH-RH-SNP-20 |
13.6 |
|
PSW-CH-RH-SNP-30 |
14.7 |
|
PSB |
3.6 |
|
PSB-CH-RH-SNP-10 |
7.9 |
|
PSB-CH-RH-SNP-20 |
10.5 |
|
PSB--CH-RH-SNP-30 |
15.4 |

Reviewer 3 Report
Please see the enclosed file.

Author Response
Responses to the Comments of Reviewer #3
Reviewer #3: This paper examines recent developments in the fabrication of eco-friendly textile fabric coatings for UV protection.
We thank the reviewer for imparting his valuable time in reviewing the manuscript
and his useful comments and recommendation for publication.
In the manuscript, I noticed two major deficiencies that need to be addressed before it is accepted for publication in coatings.
- The usage of English is far from academic and makes it difficult to be accepted at this stage.
(Authors’ Response) Thank you for reviewer comment. This has been considered in revised version and English throughout the manuscript was thoroughly revised, corrected highlighted.
- Since this is a review, the writers are required to share their own opinions and suggestions on how to move the field forward.
- (Authors’ Response) Thank you for reviewer comment. This has been considered in revised version and the all review sections discussion were strongly enriched and the results were pointed out from authors point of view.

Round 2
Reviewer 1 Report
The authors have addressed my comments properly. I recommend the publication of this work in Coatings.
Author Response
Responses to the Comments of Reviewer #1
Reviewer #1: The authors have addressed my comments properly. I recommend the publication of this work in Coatings.
We thank the reviewer for imparting his valuable time in reviewing the manuscript his recommendation for publication.

Reviewer 2 Report
The paper has considerably improved, but a few issues are missing.
- in the section on spherical nanoparticles inks, I expect a few lines on the alternative of metal-organic decomposition techniques e.g. on silver and their usage for textile coating
- text contains typos, e.g line 67 "rage"
- table on UPF is wonderful, but I expect error or uncertainty bars or at least hints in the headline about the order of magnitude
- some of the figures appear in inappropriate aspect ratio, please check.
Author Response
Responses to the Comments of Reviewer #2
Reviewer #2: The paper has considerably improved, but a few issues are missing.
We thank the reviewer for imparting his valuable time in reviewing the manuscript
and his useful comments and recommendation for publication.
- in the section on spherical nanoparticles inks, I expect a few lines on the alternative of metal-organic decomposition techniques e.g. on silver and their usage for textile coating
(Authors’ Response) Thank you for reviewer comment. This has been considered in the revised version and highlighted. Therefore, new section displaying this method and its application was inserted in revised version and highlighted. Hence, the following section and references were inserted as follows.
Inserted Section
(Page 14) On the other hand, metal–organic decomposition (MOD) ink is another promising and alternative technique for utilization in coating of textile with metal for electronic ap-plications [68]. This process implies dissolving of metal source (eg. silver) in conven-ient solvent and then evaporation of solvent [69-70]. The choice of solvent is very crit-ical, after evaporation of solvent and decomposition of organic complexes volume of MOD inks is significantly diminished [71]. This approach is highly recommended for flexible printed electronics applications [72]. Inserted References
[68] Choi, Y., Seong, K.-D., Piao, Y. Metal−Organic Decomposition ink for printed electronics. Adv. Mater. Interf. 2019, 6, 1901002.
[69] Farraj,Y., Grouchko, M., Magdassi, S. Self-reduction of a copper complex MOD ink for inkjet printing conductive patterns on plastics. Chem. Commun. 2015, 51, 1587-1590.
[70] Choi, Y.-H., Lee, J., Kim, S. J., Yeon, D.-H., Byun, Y. Highly conductive polymer-decorated Cu electrode films printed on glass substrates with novel precursor-based inks and pastes. J. Mater. Chem. 2012, 22, 3624-3631.
[71] Vaseem, M., Lee, S., Kim, J.-G., Hahn Y.-B. Silver-ethanolamine-formate complex based transparent and stable ink: Electrical assessment with microwave plasma vs thermal sintering. Chem. Eng. J. 2016,306, 796-805.
[72] Cano-Raya, C., Denchev, Z. Z., Cruz, S. F., Viana, J. C. Chemistry of solid metal-based inks and pastes for printed electronics -A review. Appl. Mater. Today 2019,15,416-430.
- text contains typos, e.g line 67 "rage"
(Authors’ Response) Thank you for reviewer comment. This has been considered in the revised version and highlighted. And rage was corrected to be range.
- table on UPF is wonderful, but I expect error or uncertainty bars or at least hints in the headline about the order of magnitude
(Authors’ Response) Thank you for reviewer comment. This has been considered in the revised version and highlighted. Hence, the general uncertainty percentage for UPF value was inserted in Table 1 headline and highlighted.
- some of the figures appear in inappropriate aspect ratio, please check.
(Authors’ Response) Thank you for reviewer comment. This has been considered in the revised version and highlighted. Hence, Figures 4 and 6 were replaced with suitable aspect ratio.
Figure 4. (a)UV–Vis spectra of the flexible transparent free-standing GLS-GRP-PVA-150 film, (b) the UV spectra region of flexible transparent free-standing GLSGRP-PVA-150 film , and (c) digital photo of the flexible transparent freestanding GLS-GRP-PVA-150 film reflecting the transparency. Reproduced with permission [28]. Copyright 2021, Elsevier.
Figure 6. (A) UV–vis spectra of AgNPs and RH-SNP-AgNPs(A), TEM images of RH-SNP, AgNPs and RH-SNP-AgNPs (B) and insets in (B) represent histogram of AgNPs. (B) Schematic diagram representing the synthesis of RH-SNP-AgNPs and their coating layer and treatment on textile fabrics. Reproduced with permission [10]. Copyright 2017, Elsevier.

Reviewer 3 Report
Dear Authors,
I have reviewed the modified manuscript. Although I noticed moderate changes in the manuscript in terms of the language, I still do not believe the manuscript is ready for publication at this stage.
This paper would benefit from more thorough proofreading. It frequently makes it difficult to follow because it has numerous grammatical mistakes, such as verb agreement. After reorganizing the material, hiring a competent English language editor can be beneficial.
Author Response
Responses to the Comments of Reviewer #3
Reviewer #3: Dear Authors, I have reviewed the modified manuscript. Although I noticed moderate changes in the manuscript in terms of the language, I still do not believe the manuscript is ready for publication at this stage.
We thank the reviewer for imparting his valuable time in reviewing the manuscript
and his useful comments and recommendation for publication.
This paper would benefit from more thorough proofreading. It frequently makes it difficult to follow because it has numerous grammatical mistakes, such as verb agreement. After reorganizing the material, hiring a competent English language editor can be beneficial.
(Authors’ Response) Thank you for reviewer comment. This has been considered in revised version and English throughout the manuscript was further thoroughly revised, corrected and highlighted. Additionally, the English was checked, revised and corrected with native English colleague.

Round 3
Reviewer 3 Report
Changes are accepted.